# Late Post-Operative Occurrence of Dentin Hypersensitivity in Adult Patients Following Allogeneic Hematopoietic Stem Cell Transplantation—A Preliminary Report

**DOI:** 10.3390/ijerph18168761

**Published:** 2021-08-19

**Authors:** Agnieszka Bogusławska-Kapała, Barbara Kochańska, Ewa Rusyan, Grzegorz Władysław Basak, Izabela Strużycka

**Affiliations:** 1Department of Comprehensive Dental Care, Medical University of Warsaw, 02-091 Warsaw, Poland; aboguslawska@wum.edu.pl; 2Department of Conservative Dentistry, Medical University of Gdańsk, 80-210 Gdańsk, Poland; barbara.kochanska@gumed.edu.pl; 3Department of Conservative Dentistry, Medical University of Warsaw, 02-091 Warsaw, Poland; rusyan@poczta.onet.pl; 4Department of Hematology, Oncology and Internal Medicine, Medical University of Warsaw, 02-091 Warsaw, Poland; grzegorz.basak@wum.edu.pl

**Keywords:** dentin hypersensitivity pain, allogeneic hematopoietic stem cell transplantation, chronic graft-versus-host disease

## Abstract

Allogeneic hematopoietic stem cell transplantation (alloHSCT) is one of the most commonly performed transplantation procedures nowadays. Despite the significant progress made in the treatment, alloHSCT is still associated with numerous complications also affecting the oral cavity. One of them is dentin hypersensitivity (DH)—a sharp, short-term pain that occurs when stimuli act on exposed dentin. Various authors point out that DH may result in a significantly lower quality of life, among other things by impeding the consumption of food as well as causing difficulties in daily oral hygiene. The aim of the study was a preliminary analysis of the incidence rate and severity of DH pain in adult patients during late period after alloHSCT. The impact of chronic graft-versus-host disease (cGvHD) and time after alloHCT were also considered. A total of 80 patients were examined. cGvHD was identified in 52 participants. The incidence rate and severity of DH pain was assessed on the basis of a questionnaire and a clinical examination. DH pain appeared a serious problem in late period after alloHSCT regardless of post-transplant time. DH primarily affected cGvHD patients. The prevention-treatment protocol for DH should be developed for this group.

## 1. Introduction

Allogeneic hematopoietic stem cell transplantation (alloHSCT) is one of the most commonly performed transplantation procedures nowadays. In this therapy a patient receives healthy stem cells from a donor to replace their own stem cells that have been destroyed by radiation or high doses of chemotherapy (conditioning) [1]. This method is used to treat many haematological diseases, including hematologic malignancies, acquired bone marrow failure syndromes, and congenital immunodeficiency [2,3]. According to the literature, the number of patients successfully treated with this method is constantly increasing [1]. Despite the significant progress made in the treatment of hematopoietic cell transplantation, alloHSCT is still associated with numerous complications that reduce the patient’s quality of life and in some cases result in the death of the recipient. Depending on the period in which they occur, these complications can be divided into early and late. Early symptoms develop during the deep bone marrow dysfunction phase lasting up to around one hundred days after transplantation. Late symptoms occur after one hundred days have passed since alloHSCT [4,5,6]. Possible observed complications include the remote toxic effect of chemo and/or conditioning radiotherapy on the body, as well as quantitative and/or functional deficiencies of the immune system or general complications arising from transplant therapy [6]. Allogeneic transplantation is associated with the high risk of graft-versus-host disease (GvHD). This disease constitutes a group of symptoms associated with the presence of immunologically competent donor cells in the organism of the recipient [7]. Chronic graft-versus-host disease (cGvHD) develops in 30% to 70% of patients between 3 and 24 months after alloHCT. According to the current diagnostic criteria, cGvHD should be diagnosed on the basis of pathognomonic symptoms of this disease and not depending on a time criterion, i.e., over 100 days from alloHCT [8]. cGvHD is currently the greatest clinical complications in patients at long term after transplantation and the most frequent cause of mortality among patients at two-year survival rate since the completion of the procedure. It was also found to significantly decrease the quality of life of the affected individuals [9]. Clinically, cGvHD resembles diseases of connective tissue related to the production of autoantibodies such as systemic lupus erythematosus or Sjögren’s syndrome with visceral widespread symptoms. The most frequently affected parts are the skin (from 70% to 100% of cGvHD cases), the oral cavity (80–90%), the liver (50–70%), the eyes (45–55%) and the esophagus (35–45%) [8]. In the course of cGvHD, an increased fibroblast proliferation and production of collagen that can lead to tissue fibrosis are observed. Moreover, patients with cGvHD are more prone to infections due to functional disorders of the immune system including delayed immunological reconstitution [8].

The literature shows that researchers are well aware of the problem of oral complications occurring up to one hundred days after alloHSCT together with their prevention and treatment [10,11]. However, there is less information on the type, frequency and severity of oral diseases occurring in adults more than one hundred days after transplantation [12]. Based on data from the literature [12,13,14], we found that the incidence rate of individual oral complications in alloHSCT patients in the late post-operative period varies, and depends, among others, on how much time has elapsed since transplantation. For example, during the deep immunological deficiency phase (between 100 days and up to 1 year following alloHSCT) and during progressive stabilization of the hematopoietic system (between one and two years following alloHSCT), the most prevalent problems are those resulting from immunosuppression and cGvHD [15]. Complications observed during the late post-operative phase (more than two years after alloHSCT) include, for example, secondary tumours [16], musculoskeletal disorders [17] and osteonecrosis of the jaw [18]. Regardless of the time elapsed since alloHSCT, patients can experience reduced salivation and xerostomia [19], dental diseases such as caries, non-carious lesions [19], as well as gingivitis and/or periodontitis [12,20]. It is essential that, the incidence of these complications is higher than in healthy subjects [13,19].

One complication that may affect the oral cavity during the late post-allotransplantation phase is dentin hypersensitivity (DH) [21]. DH is a sharp, stinging, short-term pain that occurs when thermal, chemical, dehydrative, osmotic or mechanical stimuli act on exposed dentin, and which cannot be explained by another disease or damage to dental tissue [22]. DH occurs as a result of dentin exposure with the simultaneous opening of dentinal tubules to the action of external stimuli [23]. The exposure of dentinal tubules occurs as a consequence of the loss of dental enamel/root cementum due to attrition, abfraction and abrasion of hard dental tissue or resulting from gingival recessions with or without loss of alveolar ridge [24]. The mechanism of DH pain remains poorly elucidated. The most acceptable theory explaining DH is the hydrodynamic theory [25]. It states that the flow of fluid present inside dentinal tubules is responsible for the sensation of pain. Therefore, when stimuli such as acids or cold air come into contact with the exposed dentinal surface, osmotic changes or dried dentin causes rapid dentinal fluid flow, leading to the sensation of pain [23]. Dentin sensitivity can affect single teeth, groups of teeth or entire dentition of the patient. Various authors point out that this may result in a significantly lower quality of life, among other things by impeding the consumption of food as well as causing difficulties in performing daily oral hygiene [26,27].

Increased frequency of DH in adult patients, especially during the late post alloHSCT phase, has only been addressed in one publication [21]. A need for more detailed studies in this area is, thus, essential in the future. The results obtained in the present study may be used to create algorithms for prophylactic treatment taking into account the specific general medical and dental profile of this group of patients.

The aim of this study was to provide a preliminary analysis of the incidence rate and severity of DH in patients who were at least 100 days after allogeneic hematopoietic cell transplantation, taking into account the amount of time that had elapsed since the procedure as well as the possible occurrence of cGvHD. Based on our clinical experience we supposed that DH should be a troublesome problem for this group of patients due to the provoked pain during eating, tooth brushing, speaking etc.

## 2. Material and Methods

In total, 80 adult patients took part in the study (36 women and 44 men). They were aged 18 to 66 (m ± SD = 41 ± 13; Me = 40), who had undergone alloHSCT between 100 days and 10 years previously (Table 1).

The reasons for transplantation were as follows: chronic myelogenous leukemia (CML—23 cases), acute myeloid leukemia (AML—40 cases), acute lymphoblastic leukemia (ALL—6 cases), chronic lymphocytic leukemia (CLL—2 cases), myelodysplastic syndrome (4 cases), multiple myeloma (1 case), Hodkin’s lymphoma (2 cases), Burkitt’s lymphoma (1 case), and peripheral cell lymphoma (1 case).

The chronic form of graft-versus-host disease was identified in 52 study participants The diagnosis of cGvHD was made in Department of Hematology, Oncology, and Internal Medicine, Medical University of Warsaw, Poland, according to the 2014 National Institutes of Health Consensus (the presence of at least one clinical diagnostic manifestation of chronic GVHD or at least one distinctive manifestation plus a pertinent biopsy, laboratory or other tests, evaluation by a specialist or radiographic imaging) [8].

The dental research was conducted from March 2017 to June 2019 in Department of Comprehensive Dental Care, Medical University of Warsaw, Poland. It was a part of comprehensive evaluation of stomatognathic system of the individuals who had undergone allogeneic cell transplantation in Department of Hematology, Oncology, and Internal Medicine, Medical University of Warsaw, Poland. The patients were referred to the dental department by a hematologist. At the beginning they were given detailed information about the study. Subsequently, oral and written consent of the patient was obtained for the study, as well as for the use of the results in the publication. All data included in the written consent and the questionnaire were fully anonymous. The content for the study was approved by the local bioethics committee. Then the patients were carefully examined (see below) and included to assess the presence or absence of DH. The exclusion criteria from the hypersensitivity clinical trial were complete edentulism and teeth in which the pain may have been caused by factors other than DH: dental treatment less than 4 weeks before screening or during the ongoing study, caries, defective restorations, crowns, gross periodontal disease, trauma in the past 12 months [28].

The patient population was divided into 3 groups depending on the time that had elapsed since transplantation (Table 1). In this way it was possible to consider the gradual process of hematopoietic reconstitution following alloHCT and its possible impact on oral processes. Group I comprised 31 patients who were between 100 days (3.5 months) and 10 months after transplantation (the hematopoietic system is still largely immature). Group II comprised 30 patients 12–24 months after transplantation (the hematopoietic system is stabilising). Group III (mature immune system) consisted of 19 individuals between 27 and 120 months after transplantation.

Prior to the clinical examination each patient completed a questionnaire designed to provide information on the patient’s overall health, medications taken, oral problems, perception, duration and origin of DH as well as identify risk factors for DH, including data on lifestyle, dietary factors, and oral health behavior [29]. For this purpose, the oral health assessment card created for the study was used (Appendix A). The intensity was recorded on a 101-point visual analogue scale (VAS, 0-101) with 0 indicating absence of pain, and 100 indicating the most severe pain. Cut points on the pain VAS have been recommended as: no pain (0–4 mm), mild pain (5–44 mm), moderate pain (45–74 mm), and severe pain (75–100 mm) [30].

During clinical examination each tooth was examined separately in all qualified subjects for possible pain and its severity and to determine the cause of the discomfort.

In total, 1870 teeth (Group I—680 teeth, Group II—772, Group III—418) were examined for hypersensitivity. Each qualified tooth was tested for hypersensitivity through exposure to a low temperature (cold air). The clinician applied a 1-s blast of cold air from an air-water syringe at a distance of 1 cm to the vestibular surface of the tooth, with the adjacent teeth being shielded [28]. Hypersensitivity to thermal stimuli was assessed using the three-grade scale devised by Addy, Mostaf and Newcombe [31]:1°—discomfort;2°—pain occurs after the stimulus has been activated, but disappears after cessation;3°—prolonged pain after stimulus has been withdrawn.

The examination also assessed the presence of factors that could be the cause of DH in alloHSCT patients, e.g., decreased salivation or certain stimulants.

The clinical examination was conducted by one experienced dentist—member of interdisciplinary transplantation team of Warsaw Medical University, Poland.

### Statistical Analysis

All the statistical calculations were performed using the STATISTICA data analysis software system, version 13.0 (TIBCO StatSoft Inc, StatSoft Poland, Warsaw). The results are given as the arithmetic means (m), standard deviation (SD) and median (Me). The Chi-Square non-parametric test was used to determine if there was a relationship between two variables: in this case occurring DH symptoms and suffering from cGvHD. We used the Mann–Whitney test for comparing two conditions without making the assumption that values are normally distributed and the Kruskal–Wallis Test for comparing more than two groups [32,33]. For all the calculations, statistical significance was set at *p* < 0.05.

## 3. Results

### 3.1. Self-Report DH Symptomatology

According to the interview of the 80 subjects who underwent alloHSCT, 40 (50%) reported symptoms of DH when teeth were exposed to cold and/or acidic foods and/or when exposed to air, 14 individuals (35%) reported mild sensitivity (VAS 5–44 mm), 16 (40%) reported moderate sensitivity (VAS 45–74 mm), and 10 (25%) considered their sensitivity to be severe (VAS 75–100 mm). Of the 40 patients reporting symptoms of DH, 30 (75%%) suffered from cGvHD. Patients reporting DH symptoms constituted 57.7% of all patients with cGvHD and the difference was statistically significant (*p* = 0.038). In addition, 9 out of 10 individuals who reported severe pain in their interview simultaneously suffered from cGvHD (Table 2).

### 3.2. Clinical Examination Data

#### 3.2.1. Occurrence of Dentin Hypersensitivity in the Study Group

The results of the hypersensitivity clinical examination are presented in Table 3. The data show that a high percentage of alloHCT patients in the clinical study had symptoms of DH: 58.8% (47 out of 80 subjects). The percentage of individuals with DH symptoms was comparable in Group I, II, and III. Of the 52 patients with cGvHD, clinical manifestations of DH were found in 65.4% (Table 3) while in group without cGvHD it was 50%.

#### 3.2.2. Occurrence and Intensity of Dentin Hypersensitivity in Examined Teeth

According to the data in Table 4, the physical examination revealed DH in 24.7% of all the teeth. In most cases this was grade 2 DH, i.e., acute pain that occurs when a stimulus acts on the tooth, but which rapidly abates once the stimulus is withdrawn. This pain was observed in 55.3% of all teeth with DH. Grade 1 DH, i.e., discomfort as a result of contact with a stimulus, was found in 31.9% of all teeth with DH. Grade 3 DH, i.e., severe pain that persists after cessation of the stimulus, affected 12.8% of all teeth with DH. In group I—19.9% of the teeth had grade 3 DH, compared to 8% in group II and 10.6% in group III but the differences between the groups were not statistically significant.

According to the data presented in Table 4, patients with cGvHD had a higher percentage of teeth with DH (30.3%), compared to the group without cGvHD (14.5%)—this difference was statistically significant (*p*-value < 0.001). Grade 2 DH was the predominant form of hypersensitivity experienced in clinical examination by individuals with cGvHD (57.1% of all teeth with DH in this group). On the other hand, 26.9% of the teeth in this group had grade 1 DH, while 16.0% had grade 3. The percentages of teeth with grades 1 and 2 DH in patients without cGvHD were similar and accounted for approximately 50% of all teeth with DH in this group. Only one tooth was noted as grade 3 DH.

Table 5 presents the range of incidence of DH for individual patients in terms of the percentage of teeth with clinical symptoms of hypersensitivity. According to the data from the study, in 17 (36.2%) of the 47 patients with clinical DH symptoms the condition affected between 20.1% and 50% of all teeth examined. In addition, in 16 (34%) of the 47 patients with DH symptoms, the condition affected more than 50% of all the teeth examined; in some cases, this figure was as high as 100%. However, when the presence or absence of cGvHD was taken into account, DH affected over 50% of all the teeth in 14 of 34 (41.2%) patients with cGvHD. However, only two patients in the group without cGvHD experienced DH symptoms in more than 50% of their teeth.

## 4. Discussion

The study evaluating the incidence and severity of DH in patients undergoing alloHSCT was conducted in one of the leading transplant center in Poland. The number of hematopoietic cell transplantations performed in this country averages 2000 per year [34].

During clinical interviews of patients undergoing alloHSCT, frequent complaints of pain in the form of DH were noted. A detailed analysis of the study results shows that DH is a significant oral problem in adult alloHSCT patients during the late post-operative period. Half of the patients self-reported DH symptoms that caused discomfort. At the same time, a higher percentage of individuals with DH symptoms in the clinical examination was observed. The reason for this discrepancy may be the conscious or subconscious avoidance of stimuli that could cause pain e.g., modification of eating (avoiding contact with problematic teeth, long, careful chewing), being careful when breathing, avoiding cold food, changes of toothbrushing [26]. These preventive behaviors cause the tissues to be affected by stimuli below the pain threshold.

In some individuals DH clinical symptoms was severe and affected over 50% of their teeth. It was noted that the incidence rate and severity of DH detected in the physical examination did not decrease significantly over time after transplantation. DH symptoms primarily occurred in patients with the chronic form of graft-versus-host disease. Furthermore, individuals with cGvHD were significantly more likely to report more acute DH symptoms in their medical history and during the physical examination.

Meanwhile, an analysis of the literature revealed that there are no publications dealing with the incidence of DH symptoms in patients in the late post-operative period following allogeneic hematopoietic cell transplantation. Only Alborghetti et al. observed that the majority of patients after bone marrow transplantation suffer from generalized dental tenderness. However, the authors did not provide detailed data on this issue [21].

A comparison of our results on the incidence rate of DH symptoms in alloHSCT patients with the results of studies in the literature for generally healthy persons (the general population), revealed a higher incidence of hypersensitivity in patients after transplantation. For example, in a clinical study conducted on a European population of generally healthy younger individuals (18–35 years) and employing methods similar to those used in this study, West et al. found DH symptoms in 41.9% of cases [29]. In most other adult studies, the frequency of DH was even lower and was around 25% [35,36].

In the initial assessment of DH etiological factors in the alloHSCT patients examined in the study, attention was paid to the special importance of long-term stress during the course of transplant therapy [37]. Persistent nervous tension triggers in the patient unfavorable habits (parafunctions), which, in turn, may result in pathological changes, such as the exposure of tooth roots, abrasions, abfractions, and attrition, which in turn promote the development of hypersensitivity [38].

Among the local factors that may affect the onset of DH in alloHSCT patients during the late post-operative period, consideration should be given to gingivitis and/or periodontitis [39]. Periodontopathies are diseases frequently found in patients following transplantation, especially in those affected by cGvHD [40]. During the late post-operative period alloHSCT patients often also experience significant long-term persistent saliva deficiency [13]. It is a complication that accompanies cGvHD or can also be a side effect of certain drugs (e.g., antihypertensive agents, opioids, antidepressants, tranquillizers) [41]. Saliva deficiencies may be accompanied by changes in its chemical composition, which negatively affect, for example, the remineralization of dental hard tissue [42]. The use of saliva substitutes, some of which are low in pH and/or contain citric acid, may also constitute a significant risk factor for DH [43].

Another possible cause of DH are changes in the chemical composition of saliva that may occur in patients undergoing alloHSCT. These changes are, among other things, an effect of conditioning chemotherapy [42]. For example, this procedure reduces the concentration of inorganic phosphates in saliva [42], which in turn may result in the erosion of dental hard tissue and lead to DH [44]. Another late post-operative complication in the case of alloHSCT is dysgeusia [45]. In this case, patients often change their dietary habits, including eating meals with acidic pH [46,47], which is an important risk factor for hard dental tissue loss, exposure of dentinal tubules, and symptoms of DH [48].

It is also important to remember that symptoms of DH (especially in case of severe pain) in adult alloHSCT patients should be also differentiated with other oral pain resulting from cancer and its treatment, e.g., neuropathic conditions (secondary to chemotherapy or virus infections), musculoskeletal pains (osteoporosis, arthropathy) [49,50,51]. In addition to the dental examination for the evaluation of DH, a sensory evaluation (presence of dysesthesia/paresthesia) may be important.

The detection of the presence of mix type of pain (somatic + neuropathic) or one or the other in this population, will be relevant in tailoring care protocols.

Determining the specific risk factors for DH requires further testing of a larger number of patients. We are currently conducting such studies in our clinic and the results will be published soon.

## 5. Conclusions

DH is a significant problem in adult alloHSCT patients during the late post-operative period. Half of our surveyed patients reported a history of DH, and as many as 1/4 of these patients considered this condition to be very painful (severe pain).

The clinical examination revealed DH in over 50% of patients following alloHSCT. No relationship was observed between post-transplant time and the frequency of hypersensitivity.

DH was reported in 1/4 of teeth, most of which had grade 2 hypersensitivity.

DH primarily affected cGvHD patients. Both the frequency of the condition and its severity were higher in individuals with cGvHD compared to patients without this disease.

The prevention-treatment protocol for DH should be included in the comprehensive dental care algorithm for alloHSCT patients during the late post-operative period, especially for those with cGvHD.

## Figures and Tables

**Table 1 ijerph-18-08761-t001:** Size of patient groups, depending on the time elapsed since alloHSCT, including the occurrence of cGvHD.

Group	Number of alloHSCT Patients	Time Elapsed Since alloHSCT [Months]		Age of Study Participants [Years]		Number of Patients with cGvHD	Duration of cGvHD [Months]	
	*n*	Range	m ± SD (Me)	Range	m ± SD (Me)	*n*	Range	m ± SD (Me)
I	31	3–10	5.9 ± 2.5 (6.0)	22–65	43 ± 12 (41)	18	1–7	3 ± 1.8 (3.0)
II	30	12–24	17.5 ± 4.5 (18.0)	18–66	40 ± 14 (37)	21	1–20	9.8 ± 5.3 (11.0)
III	19	27–120	45.2 ± 23.2 (36.0)	19–59	40 ± 11 (42)	13	10–80	35.9 ± 18.6 (30.0)
Total	80	3–120	19.6 ± 19.1 (14.0)	18–66	41 ± 13 (40)	52	1–80	14.0 ± 16.3 (9.0)

*p* value: not applicable; m—the arithmetic means; SD—standard deviation; Me—median; alloHSCT—allogeneic hematopoietic stem cell transplantation; cGvHD—chronic Graft versus Host Disease.

**Table 2 ijerph-18-08761-t002:** Incidence rate and severity of dentin hypersensitivity in post alloHSCT patients, including those suffering from cGvHD—questionnaire results.

Number of Patients Post alloHSCT		Number of Patients Self-Reporting Symptoms of DH	Number of Patients Self-Reporting An Amount of Pain		
			Mild Pain	Moderate Pain	Severe Pain
*n*		*n*	*n*	*n*	*n*
patients without cGvHD	28	10 ^a^	6	2	1
patients with cGvHD	52	30 ^a^	8	14	9
Total	80	40	14	16	10

Significance of differences assessed by means of U Mann–Whitney test: ^a-a^ *p*-value = 0.038, *p* < 0.05. cGvHD—chronic Graft versus Host Disease; DH—dentin hypersensitivity.

**Table 3 ijerph-18-08761-t003:** Occurrence of dentin hypersensitivity noted during the clinical examination, taking into account the time elapsed since alloHSCT and incidence of cGvHD.

Total Number of alloHSCT Patients				Patients with cGvHD			Patients without cGvHD		
Group	Number of Study Partici-Pants	Clinical Symptoms of DH		Total	Clinical Symptoms of DH		Total	Clinical Symptoms of DH	
		+	−		+	−		+	−
	*n*			*n*			*n*		
I	31	16 ^b^	15	18	11	7	13	5	8
II	30	20 ^b^	10	21	15	6	9	6	3
III	19	11 ^b^	8	13	8	5	6	3	3
Total	80	47 (58.8%)	33 (42.2%)	52	34 ^a^ (65.4%)	18 (34.6%)	28	14 ^a^ (50.0%)	14 (50.0%)

U Mann–Witney test: ^a-a^ *p*-value = 0.401, *p* > 0.05. Kruskal–Wallis Test to compare clinical symptoms of DH in I, II, III alloHSCT groups. ^b-b-b^ *p*-value = 0.733, *p* > 0.05. +: occurrence of hypersensitivity; −: no hypersensitivity; alloHSCT—allogeneic hematopoietic stem cell transplantation; cGvHD—chronic Graft versus Host Disease; DH—dentin hypersensitivity.

**Table 4 ijerph-18-08761-t004:** Number and percentage of teeth with symptoms of dentin hypersensitivity, taking into account time elapsed since alloHSCT and incidence of cGvHD.

Number of Teeth Eligible for Examination in Patients after alloHSCT		Number and Percentage of Teeth with No Symptoms of DH	Number and Percentage of Teeth with Symptoms of DH	Number and Percentage of Teeth with Different Degrees of DH	
Group	*n*	*n* (%)	*n* (%)	Degree of Sensitivity	*n* (%)
I	680	519 (76.3)	161 (23.7)	1°	41 (25.4)
				2°	88 (54.7)
				3°	32 (19.9)
II	772	585 (75.8)	187 (24.2)	1°	61 (32.6)
				2°	111 (59.4)
				3°	15 (8.0)
III	418	305 (73.0)	113 (27.0)	1°	45 (39.8)
				2°	56 (49.6)
				3°	12 (10.6)
Total	1870	1409 (75.3)	461 (24.7)	1°	147 (31.9)
				2°	255 (55.3)
				3°	59 (12.8)
Patients with cGvHD	1203	839 (69.7)	364 (30.3) ^a^	1°	98 (26.9)
				2°	208 (57.1)
				3°	58 (16.0)
Patients without cGvHD	667	570 (85.5)	97 (14.5) ^a^	1°	49 (50.5)
				2°	47 (48.4)
				3°	1 (1.1)

Significance of differences assessed by means of the chi-square test: ^a-a^ *p*-value = 0.00001, *p* < 0.01, alloHSCT—allogeneic hematopoietic stem cell transplantation; cGvHD—chronic Graft versus Host Disease; DH—dentin hypersensitivity.

**Table 5 ijerph-18-08761-t005:** Percentage of teeth with dentin hypersensitivity symptoms detected in a physical examination.

Group	Number of Persons with DH [*n*]	Number of Persons with Particular Percentage of Teeth with DH Symptoms Detected in Physical Examination			
		<10% of Teeth with DH	10.1–20% of Teeth with DH	20.1–50% of Teeth with DH	50.1–100% of Teeth with DH
I	16	3	3	4	6
II	20	1	4	9	6
III	11	1	2	4	4
Total	47	5	9	17	16
Patients with cGvHD	34	2	6	12	14
Patients without cGvHD	14	3	4	5	2

*p* value: not applicable; alloHSCT—allogeneic hematopoietic stem cell transplantation; cGvHD—chronic Graft versus Host Disease; DH—dentin hypersensitivity.

## Data Availability

The data presented in this study are available on request from the corresponding author. The data are not publicly available due to privacy.

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
