# Peer review of "Late Post-Operative Occurrence of Dentin Hypersensitivity in Adult Patients Following Allogeneic Hematopoietic Stem Cell Transplantation—A Preliminary Report"

_ijerph, 2021, doi:10.3390/ijerph18168761_

Round 1

Reviewer 1 Report

Dear Authors,

Thank you for the opportunity to review your manuscript. The aim of the presented manuscript was a preliminary analysis of the incidence rate and severity of dentin hypersensitivity in patients who were at least 100 days after allogeneic hematopoietic cell transplantation, taking into account the amount of time that had elapsed since the procedure as well as the possible occurrence of cGvHD. The topic is very interesting and obtained results are worth to be published. However, I suggest some major revisions:

  1. Introduction:

Line53: Based on our own research [10] and data from the literature [9, 11]……-could you please paraphrase the sentence. In my opinion, it would be better not used the words “our own research” especially in the introduction section.

I miss a short paragraph about the cGvHD.

Material and Methods:

Line 117-“our center”-please add the whole name of transplantation center (city and Country)

Line 126-could you add the information and citation if the scale used for the perception of DH was based on f.i. Vas scale or 4 points Likert scale or other

Please add the new paragraph called -Statistical Analysis

The choice of specific statistical tests should be confirmed in literature-please add the citation

Results

In my opinion, the tables should be inserted into the main manuscript, not into supplementary files

Discussion

Discussion is reliable and comprehensive with an actual review of the literature.

Conclusions.

The manuscript ends with the appropriate conclusions. However, could you delete the point distinction, please?

I would create so-called “Practical Application” and removed the line 289-291 to this section.

I miss a list of abbreviations.

Authors' contributions: It would be better to use only names without the tittles. The names must be replaced with abbreviations f.i. ABK….

Author Response

Dear Reviewer,

Thank you very much for your comments. Hope, we corrected all the mistakes, explained inaccuracies  and completed the data properly.

We present our responses below.

Introduction.

  • Line53: Based on our own research [10] and data from the literature [9, 11]……-could you please paraphrase the sentence. In my opinion, it would be better not used the words “our own research” especially in the introduction section.

Response: We fully accept this remark. The text has been corrected.

  • I miss a short paragraph about the cGvHD.

Response: We have included a short information about the disease in the introduction.

Material and methods.

  • Line 117 -“our center”-please add the whole name of transplantation center (city and Country)

Response: We have included information about our center: the Departament of Comprehensive Dental Care, Medical University of Warsaw, Poland and Department of Hematology, Oncology and Internal Medicine, Medical University of Warsaw, Poland.

  • Line 126 - could you add the information and citation if the scale used for the perception of DH was based on f. i. Vas scale or 4 points Likert scale or other

Response: We used VAS scale in our study. The text has been corrected.

  • Please add the new paragraph called -Statistical Analysis

Response: The text has been corrected.

  • The choice of specific statistical tests should be confirmed in literature-please add the citation

Response: We have added the relevant citation.

Results

  • In my opinion, the tables should be inserted into the main manuscript, not into supplementary files

Response: Tables are inserted in the main text now.

Discussion

  • Discussion is reliable and comprehensive with an actual review of the literature.

Conclusions.

  • The manuscript ends with the appropriate conclusions. However, could you delete the point distinction, please?

Response: The text has been corrected.

  • I would create so-called “Practical Application” and removed the line 289-291 to this section.

Response: Of course the practical application of the obtained data is very important. We would like to publish a prevention-treatment protocol for DH in a separate paper. It will be a part of the wider text referring to dental treatment in late period after HSCT. In this paper we would like to insert some more protocols taking into account the specifics of the group e. g. caries treatment, endodontics, dental surgery and prevention – based on our data and the literature.

  • I miss a list of abbreviations.

Response: A list of abbreviations is inserted in the text now.

  • Authors' contributions: It would be better to use only names without the tittles. The names must be replaced with abbreviations f.i. ABK….

Response: Authors’ contributions have been corrected.

Reviewer 2 Report

The presence of intra-oral symptomatology in patients undergoing alloHSCT is an important piece of information, that can improve and/or prevent comorbidities and discomfort, if we increase awareness among the clinicians.

However, minor and major changes need to be performed. 

In the Introduction and in the Abstract, I would advise to replace the term "today" with "nowadays" (line 14 and 33).

Line 55: replace "among other things" with "among others.

Please, clarify the sentence of the line 55. 

Once you introduce the acronym of Dentin Hypersensitivity or of graft-versus-host disease, please be consistent in using DH and cGvHD in the paper and not the full term anymore.

Line 81: no Saxon genitive in the scientific articles

Line 77: please, provide reference to the hydrodynamic theory.

Line 95: troublesome in terms of? Please clarify (e.g. in terms of quality of life, QoL, etc. the variables that you analyse in the article). 

Materials and methods:

Line 97: shift the order, "a total of 80 adult patients". How were the participants enrolled in the study? Where they contacted via telephone, selected among archival data, ... Please clarify. Where was the study conducted? The article lacks this information. 

Line 98: replace δ with SD, the usual abbreviation of standard deviation. 

Line 99: I would not repeat the acronym of alloHSCT again. It has already been defined previously. 

Line 114: explicit that group III has mature immune system.

Line 115: what do you mean with "the research was conducted"? The enrollment of the patient? the chart review?  

Line 117: reformulate the inclusion and exclusion criteria in a more clarified and structured way. 

Line 127: what do you mean with "origin"? Do you mean "site"?

Line 129: what is the authors' oral health assessment card? 

Line 130: how was the clinical examination for possible pain performed? If it is the same clinical procedure detailed in lines 138 on, you can move the description up or add a clarification, e.g. see below. 

line 151: did you test for normality first? Clarify how you used chi-square test. What did you check? 

Results:

Line 162: the difference was statistically significative (not important). Please, provide the exact p value (not just p<.05). How did you reach this statistics? chi-square of proportion between the group with cGvHD and the group without cGvHD? Please clarify. 

At a first glance, I consider the expression of the Results a bit confusing. My understood is that the % of DH of the 3.1 interview data is the self-reported and the 3.2 Clinical examination data is the one resulted from the clinical examination. If this is the correct way to interpret, I would prefer to name 3.1 as Self-report HD symptomatology (not interview data). In this way, it's better clarified that the 1st paragraph is what the patients report, in contrast to the 2nd paragraph. 

Line 169: do you refer to the data of 1st paragraph? Do not restate it here, it is referred to self-reported symptoms. You can further debate it in the discussion.

Line 171: restate the concept. It is confusing. 

Line 173: difference between which groups? the three groups of Group I, II and III? So which statistic test did you use to check for this difference? Anova test might be more appropriate when you have more than 2 groups. 

I recommend to replace the term "people" in individuals, subjects, participants, patients.

How was the cGvHD diagnosed?

Table 2: provide the exact p-value. I would recommend to perform a chi-square test for proportion and not for mean. Use ANOVA to check for significance difference in the different groups (table 2) among the cGvHD and the non cGvHD.

Table 3: provide the exact p-value . Use chi-square test for proportion, not for mean 

Table 5: Change the legend, breakdown is a confusing term, change "people". 

Discussion: I would discharge the information from 207-214. Provide reference to your statements of the 2000 per year. 

Line 216: "a detailed analysis" (not an). How can you indicate that the quality of life is reduced? The quality of life is based on a questionnaire, was it investigated? In that case, please clarify, otherwise discharge this assumption. 

Line 220: I suggest to perform a Pearson correlation, that correlated intensity of DH symptomatology and increasing time. Your statement of line 220 is intact not supported by statistics.

In the discussion, discuss further the difference between self-report DH symptomatology and clinical DH.

Provide references of the last paragraph on neuropathic pain. Unless you provide any clinical similarities between neuropathic pain symptomatology and DH, I do not think neuropathic pain should be listed in the DD.

I suggest that you provide recommendations on prevention-treatment protocol for DH (e.g. check-ups every certain months, etc), especially because you list it as a bullet point in the conclusion. 

I suggest that you add a paragraph exploring the other possible intra-oral consequences (e.g. the one that you mentioned in the introduction). This might be the only possible comparison with the literature.   

Author Response

Dear Reviewer,

Thank you very much for your comments. Hope, we corrected all the mistakes, explained inaccuracies  and completed the data properly.

We present our responses below.

  • In the Introduction and in the Abstract, I would advise to replace the term "today" with "nowadays" (line 14 and 33).

Response: The text has been corrected.

  • Line 55: replace "among other things" with "among others.

Response: The text has been corrected.

  • Please, clarify the sentence of the line 55.

Response: “Based on data from the literature [12, 13, 14], we found that the incidence rate of indi-vidual oral complications in alloHSCT patients in the late post-operative period varies, and depends, among others, on how much time has elapsed since transplantation. “ It means that some oral complications can differ depending on the time elapsed from transplantation day. For example, during the deep immunological deficiency phase and during progressive stabilization of the hematopoietic system, the most prevalent problems are those resulting from immunosuppression and cGvHD.

  • Once you introduce the acronym of Dentin Hypersensitivity or of graft-versus-host disease, please be consistent in using DH and cGvHD in the paper and not the full term anymore.

Response: The text has been corrected.

  • Line 81: no Saxon genitive in the scientific articles

Response: The text has been corrected.

  • Line 77: please, provide reference to the hydrodynamic theory.

Response: The reference has been added.

  • Line 95: troublesome in terms of? Please clarify (e.g. in terms of quality of life, QoL, etc. the variables that you analyse in the article).

Response: troublesome due to the deterioration of quality of life caused for example by provoked pain during eating, tooth brushing, speaking etc.

Materials and methods

  • Line 97: shift the order, "a total of 80 adult patients". How were the participants enrolled in the study? Where they contacted via telephone, selected among archival data, ... Please clarify. Where was the study conducted? The article lacks this information.

Response: We have completed the above data.

  • Line 98: replace δ with SD, the usual abbreviation of standard deviation.

Response: The mistake has been corrected.

  • Line 99: I would not repeat the acronym of alloHSCT again. It has already been defined previously.

Response: The mistake has been corrected.

  • Line 114: explicit that group III has mature immune system.

Response: The text has been completed.

  • Line 115: what do you mean with "the research was conducted"? The enrollment of the patient? the chart review?

Response: "the research was conducted" – we mean our research of oral cavity status of patient after alloHSCT. We hope more precise description referring to the rules of the enrolment of the patients are inserted in the text now. We have added the chart review to supplementary materials.

  • Line 117: reformulate the inclusion and exclusion criteria in a more clarified and structured way.

Response: We hope the above criteria are more transparent now.

  • Line 127: what do you mean with "origin"? Do you mean "site"?

Response: Yes, of course we mean site the pain is located.

  • Line 129: what is the authors' oral health assessment card?

Response We have added the chart review to supplementary materials.

  • Line 130: how was the clinical examination for possible pain performed? If it is the same clinical procedure detailed in lines 138 on, you can move the description up or add a clarification, e.g. see below.

Response: It is the same. We have corrected the text accordingly to your suggestion.

  • line 151: did you test for normality first? Clarify how you used chi-square test. What did you check?

Response: We asked to revise the statistics again, by another person.

In our analysis we used nonparametric tests, so they do not require the assumption that the distributions are normal.

During the analysis we used two tests.

The Chi-Square non-parametric test was used to determine if there was a relationship between two variables: in this case occurring DH symptoms and suffering cGvHD. We wanted to determine whether reporting DH symptoms depends on suffering from cGvHD. We performed the test with a contingency table with data classified according to those two variables with the total count of cases for a specific pair of categories. The null hypothesis is this case is that there is no dependency between occurring DH symptoms and suffering cGvHD.

The use of the Mann-Whitney test for comparing two conditions without making the assumption that values are normally distributed.

 In case of this work the question is:

Were patients with cGvHD in the surveys and clinical trials more likely to experience symptoms of hypersensitivity compared to those without cGvHD?

Results

  • Table 2: provide the exact p-value. I would recommend to perform a chi-square test for proportion and not for mean. Use ANOVA to check for significance difference in the different groups (table 2) among the cGvHD and the non cGvHD.

and

  • Line 162: the difference was statistically significative (not important). Please, provide the exact p value (not just p<.05). How did you reach this statistics? chi-square of proportion between the group with cGvHD and the group without cGvHD? Please clarify.

Response: The p-value is 0.038. The result is significant at p < .05. Using the Mann-Whitney test we concluded that patients with cGvHD had a history of dentine hypersensitivity symptoms significantly more often than patients without cGvHD.

  • At a first glance, I consider the expression of the Results a bit confusing. My understood is that the % of DH of the 3.1 interview data is the self-reported and the 3.2 Clinical examination data is the one resulted from the clinical examination. If this is the correct way to interpret, I would prefer to name 3.1 as Self-report HD symptomatology (not interview data). In this way, it's better clarified that the 1st paragraph is what the patients report, in contrast to the 2nd paragraph.

Response: We have changed 3.1. name to Self-report DH symptomatology

  • Line 169: do you refer to the data of 1st paragraph? Do not restate it here, it is referred to self-reported symptoms. You can further debate it in the discussion.

Response: We discharged this statement from line 169.

  • Line 171: restate the concept. It is confusing.

Response: We have restated the concept. We meant that the percentage of individuals with DH symptoms was comparable in Group I, II and III.

  • Line 173: difference between which groups? the three groups of Group I, II and III? So which statistic test did you use to check for this difference? Anova test might be more appropriate when you have more than 2 groups.

Response: Here we used Kruskal-Wallis Test for comparing more than two groups. This is also a nonparametric test. The result was that the p-value is 0.733. Thus, the result is not significant at p < .05.

  • I recommend to replace the term "people" in individuals, subjects, participants, patients.

Response: The mistake has been corrected.

  • How was the cGvHD diagnosed?

Response: The diagnosis of cGvHD was made in Department of Hematology, Oncology and Internal Medicine, Medical University of Warsaw, Poland, accordingly to the indications of the 2014 National Institutes of Health Consensus (the presence of at least one clinical diagnostic manifestation of chronic GVHD or at least one distinctive manifestation plus a pertinent biopsy, laboratory or other tests, evaluation by a specialist or radiographic imaging).

  • Table 5: Change the legend, breakdown is a confusing term, change "people".

Response: Table has been changed.

  • Discussion: I would discharge the information from 207-214. Provide reference to your statements of the 2000 per year.

Response: We have discharged detailed information and provided a relevant reference to 2000/year.

  • Line 216: "a detailed analysis" (not an). How can you indicate that the quality of life is reduced? The quality of life is based on a questionnaire, was it investigated? In that case, please clarify, otherwise discharge this assumption.

Response: We have corrected the mistake. We have discharged the assumption about QL.

  • Table 3: provide the exact p-value . Use chi-square test for proportion, not for mean

and

  • Line 220: I suggest to perform a Pearson correlation, that correlated intensity of DH symptomatology and increasing time. Your statement of line 220 is intact not supported by statistics

Response:

In this part of our study we wanted to answer following questions. For that reason we used the Chi-Square non-parametric test.

- Do clinical signs of hypersensitivity depend on time after alloHSCT among people suffering from cGvHD?

The p value was 0.999. Clinical symptoms of hypersensitivity are independent of the time elapsed after alloHSCT with a significance level of 0.01 among people with cGvHD.

- Do clinical signs of hypersensitivity depend on time after alloHSCT in people not suffering from cGvHD?

The p value was 0.423. Clinical signs of hypersensitivity are independent of the time elapsed after alloHSCT with a significance level of 0.01 among those without cGvHD.

  • In the discussion, discuss further the difference between self-report DH symptomatology and clinical DH.

Response: We discuss the problem in the text.

  • Provide references of the last paragraph on neuropathic pain. Unless you provide any clinical similarities between neuropathic pain symptomatology and DH, I do not think neuropathic pain should be listed in the DD.

Response: References have been provided. 

  • I suggest that you provide recommendations on prevention-treatment protocol for DH (e.g. check-ups every certain months, etc), especially because you list it as a bullet point in the conclusion.

Response: Of course the practical application of the obtained data is very important. We would like to publish a prevention-treatment protocol for DH in a separate paper. It will be a part of the wider text referring to dental treatment in late period after HSCT. In this paper we would like to insert some more protocols taking into account the specifics of the group e. g. caries treatment, DH, endodontics, dental surgery and prevention – based on our data and the literature.

  • I suggest that you add a paragraph exploring the other possible intra-oral consequences (e.g. the one that you mentioned in the introduction). This might be the only possible comparison with the literature.

Response: Thank you very much for this suggestion. Based on our experience we can conclude that the problem of oral late HSCT complications, their prevention and treatment is very important. Unfortunately underestimated. But it is also wide subject covering not only dental issue but other orofacial structures. Our intention is to publish a series of articles in this area.

Reviewer 3 Report

This study describes a clinical investigation of dentin hypersensitivity in patients following allogeneic hematopoietic stem cell transplantation. This topic is of great interest, as the affected patients may suffer from this dental problem additionally to their general state of health.

The investigation is well designed and presented, although in the material and method section the qualification and calibration of the investigators should be mentioned.

The results are clearly presented and the conclusions drawn by the authors are supported by the data. I would only suggest to state in the conclusion section, if there are any plans to report more data on this topic, as the authors state in the title of the manuscript "preliminary report".

Author Response

Dear Reviewer,

Thank you very much for your comments. Hope, we corrected all the mistakes, explained inaccuracies  and completed the data properly.

We present our responses below.

  • This study describes a clinical investigation of dentin hypersensitivity in patients following allogeneic hematopoietic stem cell transplantation. This topic is of great interest, as the affected patients may suffer from this dental problem additionally to their general state of health.

Response: Thank you very much for this comment. It is also our experience that patients in late HSCT period suffer from several general and oral problems.  So we would like to publish a series of articles in this area based on our data and the literature.

  • The investigation is well designed and presented, although in the material and method section the qualification and calibration of the investigators should be mentioned.

Response: Only one investigator was involved in the study. The information is in the text now.

  • The results are clearly presented and the conclusions drawn by the authors are supported by the data. I would only suggest to state in the conclusion section, if there are any plans to report more data on this topic, as the authors state in the title of the manuscript "preliminary report".

Response: Yes, we would like to publish an article based on higher group, with quality of life investigation.

Round 2

Reviewer 1 Report

All my comments have been taken into account. Thank you.

Author Response

Dear Reviewer,
I would like to thank you for the very valuable comments in your review, which contributed to the increase of the value of our article.
Yours sincerely

Izabela Strużycka

Reviewer 2 Report

I compliment the authors with the new version of the manuscript. The revision allowed it to increase in soundness and quality.

Few caveats need to be modified:

Introduction: 

  • line 43 : the references [4,5,6] need to be presented as [4-6]
  • line 50: "according to the current diagnostic criteria" would probably fit better than "current views", as long as  it is based on diagnostic criteria.
  • line 52: "not DEPENDING ON a time criterion"   
  • line 53: "the greatest clinical  COMPLICATIONS" (instead of "problem")
  • line 53: please replace "in late period" with "at long term"  
  • line 54: replace "among patients who lived for 2 years since the completion of the procedure" with "at two year survival rate".
  • line 55: "it was also found to significantly decrease the quality of life..."
  • line 57: replace "and the symptoms can be observed in every organ" with "with visceral widespread symptoms". 
  • provide a reference to the sentence line 58-60
  • line 60: "an increased fibroblast proliferation"

Methods: 

  • line 111: please, refer to standard deviation as SD as you indicate in the table. It is way more common and understandable
  • line 159: App 1 is the Suppl. 1? In case, please modify it. 
  • line 160-161: just put as followed. "on a 101-point visual analogue scale (VAS, 0-101) with 0 indicating absence of pain, and 100 indicating the most severe pain" and delete the sentence from "in this scale the amount [...] to an extreme". Moreover, I do not find the information that you provided in the reference by Younger et al. to support the sentence of lines 161-163. The article by Younger et al provided with VAS, but not the cut-off. Please, provide the correct reference.

Stat analysis: 

  • line 185: "suffering FROM cGvHD".
  • line 185: "We USED Mann-Whitney U test for"
  • again, did you check if the sample was normally distributed or you just  implied that it was not? Indeed, in statistics, parametric tests are to be preferred. So, I encourage you to assess for normality. In case it is normally distributed, you cannot use these tests. If it is not normally distributed, in that case you can decide to utilize non-parametric test or to transform the data to normally distributed. Please, check for normality. 

Results:

  • table 2: in the footnote, "Mann-Whitney U test" (also table 3); "a-b" needs to be superscript (also table 4). Please, move the line referred to the Total in the table at the bottom, to be consistent with the format you used for Table 1 and Table 3.
  • table 3: same note referred to the superscript a and b in the footnote. Kruskal-Wallis Test "to compare " 
  • line 234: "according to the data in Table 4" instead of "as the data presented in table 4 show"
  • line 236: when a p-value is < .001, you just indicate p-value <0.001
  • line 239: "had grade 3" instead of "were grade 3". 
  • line 240: remove the comma after cGvHD; replace "amounted to around" with "accounted for approximately 50%"
  • table 4: be consistent with the space between the Legend and the table (you did not insert any spaces in the previous tables). Remove all the % after the numbers. Indeed, if you indicate n(%), this means that all the numbers in parenthesis below are %, without having to clarify it again. 
  • line 267: "was taken into account, instead of "is taken into account"
  • Table 5: "Percentage of teeth with dentin hypersensitivity symptoms detected during the physical examination". Remove all the articles from the table (the number, the percentage, etc)

Discussion: 

  • line 285-286: a higher percentage [...] WAS observed.
  • line 286: may be (instead of could be)
  • line 289: "these preventive BEHAVIORS 
  • line 336: "neuropathic conditions (secondary to chemotherapy or virus infections), musculoskeletal pain" 
  • line 337: [50-52]
  • line 337-339: why is it important to clarify if the neuropathic pain was present before or was secondary to the treatment? Would the management change?  

Author Response

Dear Reviewer,

Thank you very much for your insightful review and valuable comments, which will certainly raise the substantive value of our article.

Introduction: 

  • line 43 : the references [4,5,6] need to be presented as [4-6]

Response: The text has been corrected.

  • line 50: "according to the current diagnostic criteria" would probably fit better than "current views", as long as  it is based on diagnostic criteria.

Response: The text has been corrected.

  • line 52: "not DEPENDING ON a time criterion"  

Response: The mistake has been corrected.

  • line 53: "the greatest clinical  COMPLICATIONS" (instead of "problem")

Response: The text has been corrected.

  • line 53: please replace "in late period" with "at long term"  

Response: The text has been corrected.

  • line 54: replace "among patients who lived for 2 years since the completion of the procedure" with "at two year survival rate".

Response: The text has been corrected.

  • line 55: "it was also found to significantly decrease the quality of life..."

Response: The text has been corrected.

  • line 57: replace "and the symptoms can be observed in every organ" with "with visceral widespread symptoms". 

Response: The text has been corrected.

  • provide a reference to the sentence line 58-60

Response: The reference has been provided.

  • line 60: "an increased fibroblast proliferation"

Response: The text has been corrected.

Methods: 

  • line 111: please, refer to standard deviation as SD as you indicate in the table. It is way more common and understandable

Response: The text has been corrected.

  • line 159: App 1 is the Suppl. 1? In case, please modify it. 

Response: App 1 means the Suppl. 1. The text has been corrected.

  • line 160-161: just put as followed. "on a 101-point visual analogue scale (VAS, 0-101) with 0 indicating absence of pain, and 100 indicating the most severe pain" and delete the sentence from "in this scale the amount [...] to an extreme". Moreover, I do not find the information that you provided in the reference by Younger et al. to support the sentence of lines 161-163. The article by Younger et al provided with VAS, but not the cut-off. Please, provide the correct reference.

Response: The text has been corrected and correct reference provided.

Stat analysis: 

  • line 185: "suffering FROM cGvHD".

Response: The mistake has been corrected.

  • line 185: "We USED Mann-Whitney U test for"

Response: The mistake has been corrected.

  • again, did you check if the sample was normally distributed or you just  implied that it was not? Indeed, in statistics, parametric tests are to be preferred. So, I encourage you to assess for normality. In case it is normally distributed, you cannot use these tests. If it is not normally distributed, in that case you can decide to utilize non-parametric test or to transform the data to normally distributed. Please, check for normality. 

Response: The normality of the data was checked. For this purpose, we used the Shapiro-Wilk test of normality (the same results we obtained from the Kolmogorov-Smirnov test). In every case we checked, the p-value was well below 0.01.

Results:

  • table 2: in the footnote, "Mann-Whitney U test" (also table 3); "a-b" needs to be superscript (also table 4). Please, move the line referred to the Total in the table at the bottom, to be consistent with the format you used for Table 1 and Table 3.

Response: The tables have been corrected.

  • table 3: same note referred to the superscript a and b in the footnote. Kruskal-Wallis Test "to compare " 

Response: The table has been corrected.

  • line 234: "according to the data in Table 4" instead of "as the data presented in table 4 show"

Response: The table has been corrected.

  • line 236: when a p-value is < .001, you just indicate p-value <0.001

Response: The text has been corrected.

  • line 239: "had grade 3" instead of "were grade 3".

Response: The mistake has been corrected.

  • line 240: remove the comma after cGvHD; replace "amounted to around" with "accounted for approximately 50%"

Response: The text has been corrected.

  • table 4: be consistent with the space between the Legend and the table (you did not insert any spaces in the previous tables). Remove all the % after the numbers. Indeed, if you indicate n(%), this means that all the numbers in parenthesis below are %, without having to clarify it again. 

Response: The text has been corrected.

  • line 267: "was taken into account, instead of "is taken into account"

Response: The mistake has been corrected.

  • Table 5: "Percentage of teeth with dentin hypersensitivity symptoms detected during the physical examination". Remove all the articles from the table (the number, the percentage, etc)

Response: The text has been corrected.

Discussion: 

  • line 285-286: a higher percentage [...] WAS observed.

Response: The mistake has been corrected.

  • line 286: may be (instead of could be)

Response: The mistake has been corrected.

  • line 289: "these preventive BEHAVIORS 

Response: The mistake has been corrected.

  • line 336: "neuropathic conditions (secondary to chemotherapy or virus infections), musculoskeletal pain" 

Response: The text has been corrected.

  • line 337: [50-52]

Response: The mistake has been corrected.

  • line 337-339: why is it important to clarify if the neuropathic pain was present before or was secondary to the treatment? Would the management change?  

Response: Thank you for this comment. Of course the cancer management would not change. We have removed this unclear sentence.